# On Entropic Learning from Noisy Time Series in the Small Data Regime

**DOI:** 10.3390/e26070553

**Published:** 2024-06-28

**Authors:** Davide Bassetti, Lukáš Pospíšil, Illia Horenko

**Affiliations:** 1Faculty of Mathematics, RPTU Kaiserslautern-Landau, Gottlieb-Daimler-Str. 48, 67663 Kaiserslautern, Germany; 2Department of Mathematics, Faculty of Civil Engineering, VŠB-TUO, Ludvika Podeste 1875/17, 708 33 Ostrava, Czech Republic; lukas.pospisil@vsb.cz

**Keywords:** small data, entropic AI, Markov processes, machine learning, time series

## Abstract

In this work, we present a novel methodology for performing the supervised classification of time-ordered noisy data; we call this methodology Entropic Sparse Probabilistic Approximation with Markov regularization (eSPA-Markov). It is an extension of entropic learning methodologies, allowing the simultaneous learning of segmentation patterns, entropy-optimal feature space discretizations, and Bayesian classification rules. We prove the conditions for the existence and uniqueness of the learning problem solution and propose a one-shot numerical learning algorithm that—in the leading order—scales linearly in dimension. We show how this technique can be used for the computationally scalable identification of persistent (metastable) regime affiliations and regime switches from high-dimensional non-stationary and noisy time series, i.e., when the size of the data statistics is small compared to their dimensionality and when the noise variance is larger than the variance in the signal. We demonstrate its performance on a set of toy learning problems, comparing eSPA-Markov to state-of-the-art techniques, including deep learning and random forests. We show how this technique can be used for the analysis of noisy time series from DNA and RNA Nanopore sequencing.

## 1. Introduction

Learning from time-ordered data represents an important problem in machine learning and AI. One of the first approaches used in time series modeling was to predict future values of time series from previous ones, using (auto) regressive [1,2] or moving average models [3,4]. Later on, it was shown how learning from such data can benefit from the identification of latent states and their sequences, performed through hidden Markov models (HMMs) [5] or Markov switch models [6,7]. The usage and applicability of these tools was greatly enhanced by the introduction of computationally scalable and efficient strategies for solution (e.g., the Baum–Welch algorithm [8]) and inference of the most likely sequences of latent states (Viterbi paths) [9].

Recently, deep learning methods became increasingly popular in time series analysis applications. Deep neural networks (DNNs) gained the ability to capture temporal information thanks to several advancements such as the use of recurrent connectivity [10], convolution in time [11], memory [12], or the attention layers [13] in transformers models [14]. Those factors have also significantly increased the capabilities of deep learning in learning from text and sequence data. Nevertheless, despite their success, explainability and computational cost scaling remain very challenging issues in DNN applications [15,16,17]. This is particularly relevant for models that operate in the so-called overparametrized regime [18,19,20] and on time series [21].

A mathematical framework of approximate entropic learning that was introduced very recently [22,23,24,25,26,27,28,29,30] promises to provide robust, computationally scalable, and explainable ways of machine learning and AI in the so-called “small data” regime, when the underlying learning task is highly underdetermined due to a large problem dimension and relatively small data statistics size. Entropic learning methods have been successfully applied to various domain disciplines, demonstrating superior performance on a range of long-standing “small data learning” problems in weather/climate research (like the intra-seasonal El Nino prediction) [24,27,30], in material sciences (like detection of material inhomogeneities from magnetic imaging data) [25], in biomedicine (like learning from omics data [22,23,25,26,30], in processing of ultra-noisy CT images in the ultra-low radiation regime [29]), and in economics (like prediction of stock over- and under- performance based on short and non-stationary company data time series) [28].

In the following, we provide more details on the entropic learning and present a novel extension of this methodology that allows learning from short and noisy (time-)ordered data.

### 1.1. Denoising Time Series

One common task in time series analysis is the removal of noise. Let us consider, as an example, the application of denoising a signal obtained from a Markov process that alternates between different states. In each time point, a value is sampled from a state-dependent probability distribution. This is illustrated in Figure 1A, where two normal distributions are used as probability distributions for each state, each possessing its own mean, but having the same variance.

Let us consider the identification of the state at a given time by using techniques that do not intrinsically possess a notion of order or time, for example, k-means clustering [31,32]. Using this technique involves calculating mean values and the assignments of each point to clusters is given by inspecting the Euclidean distance. Note how the same answer would be obtained for any permutation of the input data. That is because k-means is invariant with respect to ordering of the data points and ignores the time information. The assignment can be performed on new data by calculating the optimal parameters from a portion of recording where labels are given, and applying them to a portion without labels. For situations with low levels of noise and with mean values of the state-dependent distributions that are sufficiently far apart (Figure 1A, bottom left panel), k-means clustering can correctly identify the states through the affiliation of each data point to each cluster. This is not surprising, since the data are inherently clustered, as each state is characterized by a specific distribution of the data points, and the distributions can be separated. However, lower signal-to-noise ratios or overlapping mean values can rapidly lead to degradation in performance, as the clusters cannot be easily separated anymore. As illustrated in Figure 1A, in the bottom right panel, it is easy to understand why in this case, i.e., with large standard deviation, points belonging to the red state may be mistaken for points emitted in the blue state, by virtue of their mean being closer to 1 than to 2, and vice versa. Even incorporating the information about the labels, by using as assignment the labels of the training dataset, would not improve the performance on the test dataset of such a classifier.

#### 1.1.1. FEMH1

By including an additional term to the optimization problem solved while finding the k-means solution, it is possible to introduce a penalty that considers a notion of smoothness, such as the H1 seminorm [33,34], when solving the problem. This method, which is called FEMH1 since it is based on the finite elements method [33], consists of finding the optimal parameters $\Gamma^{*}$ and $C^{*}$ that, given a data matrix $X\in R^{D\times \;T}$ with *D* features and *T* data points, minimize the following functional:(1)$$\begin{array}{c} \begin{array}{c} L_{FEMH1}=\sum_{t=1}^{T}\sum_{k=1}^{K}\sum_{d=1}^{D}\Gamma_{k,t}{\left(X_{d,t}-C_{d,k}\right)}^{2}+\varepsilon^{2}\sum_{k=1}^{K}\sum_{t=1}^{T-1}{\left(\Gamma_{k,t+1}-\Gamma_{k,t}\right)}^{2} \end{array} \end{array}$$
under the condition that Γ is a stochastic matrix, i.e.,: (2)$$\forall t,k:0\leq \Gamma_{k,t}\leq 1,$$
(3)$$\forall t:\sum_{k=1}^{K}\Gamma_{k,t}=1,$$
where $C\in R^{D,K}$ is a matrix collecting a set of *D*-dimensional points (one for each cluster), $\Gamma \in {\left[0,1\right]}^{K\times T}$ is a matrix expressing the affiliation probability of each data point to any of the *K* clusters, and ε is a parameter which controls the relative weight of each term of the problem (discretization vs. smoothness) [35]. Problem (1) can be solved iteratively by finding the optimal values of *C* and Γ while keeping, respectively, Γ and *C* fixed. This formulation retains the simplicity of the k-means algorithm, together with its efficiency in terms of computational complexity, and is able to achieve impressive levels of denoising at very small signal-to-noise ratios, while the main computational cost of this algorithm is the quadratic programming (QP) problem introduced when solving for Γ [36], an efficient solver for this step has been formulated by leveraging the structure of the specific problem, and is described in [34].

#### 1.1.2. Overlapping Means

In the previous section, we discussed the inclusion of a notion of smoothness to bias the solution towards one that displays less pronounced regime switching. This, however, does not grant a good performance in case of overlapping means. Indeed, if the state-dependent distributions possess the same mean but a different variance, the technique above will not necessarily be able to correctly assign data points to the right cluster. This is illustrated in Figure 1B, where the two state-dependent distributions are selected to be normal distributions, with the same mean and different standard deviation. Note how the second dimension is uniformly distributed, and a scatter plot of the first two dimensions (right panel) leads to a problem that is not easily separable, even including smoothness.

A way to obviate to this issue can be the use of a more general set of distributions instead of centroids (which can be considered as a Delta distribution for each cluster), and of likelihood instead of Euclidean distances, to assign points, in a similar way to Gaussian mixture models (GMMs) [37,38]. GMMs involve fitting multivariate Gaussian distributions to data, by using the expectation maximization algorithm. The result of using this method is the description of the data using a mixture of distributions with the highest likelihood. The advantage introduced by this approach is that it allows, e.g., to recognize clusters having overlapping means but different variance. The price, however, is that (i) the technique becomes a parametric method, and thus we must be able to confidently assume that the data are indeed classifiable using a mixture of Gaussian distributions [39], and (ii) the use of multivariate Gaussian distributions can translate to an elevated computational cost in the case of data with high dimensionality, which is often the case in practical applications and most notably in natural sciences.

### 1.2. Small Data and Entropic Learning

#### 1.2.1. Small Data

A disadvantage of neural networks is that, in order to learn robust functions, they tend to require a considerable amount of training data, which lead to coining the term “Big Data”, directly referring to the need for a large amount of training samples [40]. It is to be noted that the needed amount of training data directly depends on the dimensionality of the problem at hand. The opposite of “Big Data” is “Small Data”, which instead refers to a situation where the number of dimensions (or features) is elevated, and/or the amount of data points is low [41]. A concrete example is from the field of -omics, where technical advancements rendered possible to measure from a small population an extremely rich set of features, and often the dimensionality can exceed the number of instances [42]. This situation is traditionally challenging, as in induces overfitting, where models can learn perfectly what they have seen during training, but are not able to generalize to novel data [43].

#### 1.2.2. High Dimensionality

In many cases, the dimensions that are actually relevant with respect to the learning task are often only a subset or a subspace of the original ones. Redundant or irrelevant measures can constitute a large part of the training data, which can contribute to the cost of the fitting without providing advantages; therefore, their exclusion constitutes a benefit. Common machine learning techniques do not include the possibility to identify and select the relevant dimensions. Traditionally, the subset or the subspace of relevant feature is selected before the training in a separate step in which other tools, such as principal component analysis (PCA) [44], tSNE [45], or UMAP [46], are used for preprocessing [47]. It is important to notice, however, that those techniques (in their original formulation) are unsupervised, and thus are not “aware” of the learning problem that will subsequently receive and use their output. PCA, for example, will produce an approximation of the data by projection on a lower dimensional (linear) manifold, such that most of the variance of the original data is retained in the approximation. This is not necessarily the projection that (i) better separates the label, or (ii) better represents the data, given the linearity of the manifold. Similar arguments can be used with regards to unsupervised nonlinear dimensionality reduction methods used as preprocessing steps. This fact is illustrated in Figure 1C, where the ideal nonlinear manifold would allow the simple separation of the two classes, but taking the first principal component would instead lead to inability to do so.

Furthermore, in case of projection-based dimensionality reduction, it does not guarantee the simplification of the future data collection process, as all dimensions may be needed to perform the projection. Instead, the dimensionality reduction methods that return a subset of the features allow the use of further data, which only contain measurements in the relevant subset of features to be processed by the model. A variation that maximizes the separability between classes, linear discriminant analysis (LDA) [48], similarly does not take into account the entire learning problem and is based on a linear manifold, which may not, in general, be sufficient. Furthermore, LDA does not typically provide good results when the means of the clusters are overlapping. Moreover, both PCA and LDA can be negatively affected in terms of the performance in the conditions of small sample size with respect to the number of feature dimensions [49].

#### 1.2.3. Entropic Learning

Entropic learning refers to a mathematical framework proposing a solution to the aforementioned problems, as it has shown the ability to successfully operate in the small data condition, outperforming state-of-the-art techniques in terms of both performance and training cost, and has been applied in various disciplines, including climate science [50,51], financial applications [28], medical imaging [29], natural sciences [52], and computer science (application to the reduction in weights in neural networks [53]). The central idea of the framework is to formulate the learning task as a single holistic mathematical problem that includes all of the standard pipeline for machine learning, including feature selection, discretization, and either classification [22] or regression [27], thus removing the need for external dimensionality reduction. The classification variant (called eSPA+) uses as input a data matrix $X\in R^{D\times T}$, where *D* and *T* refer to the number of features and instances, respectively, and a label probability matrix $\Pi \in {\left[0,1\right]}^{M\times T}$, where *M* is the number of possible labels. To fit the model, we seek the parameters $S^{*},\Gamma^{*},W^{*},\Lambda^{*}$ that minimize the following functional: (4)$$\begin{array}{c} \begin{array}{c} \begin{array}{cc} L_{\mathrm{eSPA}+} & =\frac{1}{T}\sum_{d=1}^{D}W_{d}\sum_{t=1}^{T}{\left(X_{d,t}-{\left\{S\Gamma \right\}}_{d,t}\right)}^{2}\\ \; & +\varepsilon_{E}\sum_{d=1}^{D}W_{d}\log\left(W_{d}\right)\\ \; & -\frac{\varepsilon_{CL}}{T}\sum_{m=1}^{M}\sum_{t=1}^{T}\Pi_{m,t}\log\left(\sum_{k=1}^{K}\Lambda_{m,k}\Gamma_{k,t}\right), \end{array} \end{array} \end{array}$$
such that: (5)$$\forall t,k:0\leq \Gamma_{k,t}\leq 1,$$
(6)$$\forall t:\sum_{k=1}^{K}\Gamma_{k,t}=1,$$
(7)$$\forall d:0\leq W_{d}\leq 1$$
(8)$$\sum_{d=1}^{D}W_{d}=1,$$
(9)$$\forall m,k:0\leq \Lambda_{m,k}\leq 1$$
(10)$$\forall k:\sum_{m=1}^{M}\Lambda_{m,k}=1$$

In Problem (4), each parameter can provide insight on the operation of the model and has a specific meaning, that will be described in the next paragraphs. Matrix $\Gamma \in {\left[0,1\right]}^{K,T}$, similarly to its role in Problem (1), is a stochastic matrix representing the probability of each data point to belong to each of the *K* clusters. The number of clusters *K* is an hyperparameter that can be learned from the data by using cross-validation. $S\in R^{D\times K}$ describes the position of the centroid for each cluster, $\Lambda \in {\left[0,1\right]}^{M\times K}$ is a stochastic classification matrix that indicates, for each cluster, what is the probability of assuming each label, and $W\in {\left[0,1\right]}^{D}$ is a probability vector indicating the contribution of each dimension. As with k-means, the space is implicitly discretized by a Voronoi tessellation, with the addition that each cell possesses a label probability distribution (depicted in Figure 1D, where the color represents the label probability). The introduction of *W*, which is regularized using entropy (hence the name, Entropic learning), allows feature selection to be included in the solution of the problem. Thus, this formulation not only effectively allows combining discretization, dimensionality reduction and classification in a single problem, but also guarantees that each step is optimal with respect to the problem in its entirety. As with FEMH1, we can solve Problem (4) by iteratively solving the problem with respect to each single variable, considering the others fixed, until convergence is reached.

Note that this is an entirely non-parametric method, i.e., we do not rely on any assumption of linearity as the optimal combination of discretization/dimensionality reduction/classification is learned from the data, “from the perspective of” the learning problem. This means, practically, that, for example, a dimension with a lot of variance but which is irrelevant to the learning problem will (rightfully) not be considered. The label probabilities for a data point are assigned through the use of Voronoi cells which can tessellate the feature space in any arbitrary possible way, effectively allowing nonlinear and non-dyadic relationships to be captured (as illustrated in panel D of Figure 1). The sensitivity to perturbations of this type of discretization can be precisely formulated in mathematical terms, through the definition of minimal adversarial distance [54].

This method proved to be able to operate in the small data regime [22], but it does not include a notion of time or ordering, and can suffer from the same limitation as FEMH1 with respect to overlapping means. In this work, we combine the two methodologies (FEMH1 with eSPA+), in order to expand the application to supervised classification problems to ordered data. This allows us to mutually mitigate the main weaknesses of the two methods, since the formulation of FEMH1 did not allow solving supervised problems, while eSPA+ does not allow to include time ordering information. At the same time, we endow each cluster with a probability distribution, so that the affiliation of new data points can be calculated using the likelihood instead of the Euclidean distance.

## 2. Materials and Methods

### 2.1. Mathematical Formulation of eSPA-Markov

As mentioned in Section 1, the method we will describe, which we name eSPA-Markov, originates from the confluence of eSPA [22,23,24] and FEMH1 [33,34]. Let us define as input a data matrix $X\in R^{D\times T}$, where *D* is the number of dimensions, or features, and *T* is the size of the statistics, i.e., the number of sampled points. Since we are dealing with a supervised classification problem, we also accept as input a matrix of label probabilities $\Pi \in {\left[0,1\right]}^{M\times T}$, where *M* is the number of possible labels. Note that Π is a matrix of probabilities, and as such it is assumed that each entry is a number between zero and one, and that each column of this matrix sums to one. Effectively, each column represents the discrete probability distribution over the possible *M* labels for a specific data point. This notation generalizes one-hot encoding, which is the special case where $\forall m,t:\Pi_{m,t}\in \left\{0,1\right\}$ with m=1,2,…,M and t=1,2,…,T. In terms of our algorithm, there is no difference between having one-hot-encoded labels and probabilities.

As with eSPA, we would like to simultaneously find the optimal discretization of the input data, and assign to each of those cells a label probability distribution that is the best in term of Kullback–Leibler divergence between the reconstruction and the true labels. Furthermore, we also simultaneously select the most relevant features through the use of the vector *W*, which is regularized through its entropy. The functional to be minimized is the following: (11)$$\begin{array}{c} \begin{array}{c} \begin{array}{cc} L_{\mathrm{eSPA}\text{-}\mathrm{Markov}} & =\frac{\varepsilon_{L}}{T}\sum_{d=1}^{D}\sum_{k=1}^{K}\sum_{t=1}^{T}W_{d}\Gamma_{k,t}L\left(X_{d,t},\theta_{d,k}\right)\\ \; & +\varepsilon_{E}\sum_{d=1}^{D}W_{d}\log\left(W_{d}\right)\\ \; & -\frac{\varepsilon_{CL}}{T}\sum_{m=1}^{M}\sum_{t=1}^{T}\Pi_{m,t}\log\left(\sum_{k=1}^{K}\Lambda_{m,k}\Gamma_{k,t}\right)\\ \; & +\frac{1}{T}\sum_{k_{1}=1}^{K}\sum_{k_{2}=1}^{K}\sum_{t=1}^{T-1}{\left(\Gamma_{k_{1},t+1}-P_{k_{1},k_{2}}\Gamma_{k_{2},t}\right)}^{2}, \end{array} \end{array} \end{array}$$
which is subject to the following constraints: (5)–(10).

In Equation (11), *L* represents a generic loss function, which can be, e.g., the Euclidean distance between the original data points and the centroid $\theta_{k}$ for a given cluster *k*. Without a loss of generality, we will focus on the case where the loss is calculated using ℓ(x,μ,σ), which is the negative 1 dimensional Gaussian log likelihood of a data point (*x*), and $\theta_{d,k}$ are the parameters $\mu_{d,k},\sigma_{d,k}$ of the centroid-dependent distributions. Doing so results in the following formulation: (12)$$\begin{array}{c} \begin{array}{c} \begin{array}{cc} L_{\mathrm{eSPA}\text{-}\mathrm{Markov}} & =\frac{\varepsilon_{L}}{T}\sum_{d=1}^{D}\sum_{k=1}^{K}\sum_{t=1}^{T}W_{d}\Gamma_{k,t}\ell \left(X_{d,t},\mu_{d,k},\sigma_{d,k}\right)\\ \; & +\varepsilon_{E}\sum_{d=1}^{D}W_{d}\log\left(W_{d}\right)\\ \; & -\frac{\varepsilon_{CL}}{T}\sum_{m=1}^{M}\sum_{t=1}^{T}\Pi_{m,t}\log\left(\sum_{k=1}^{K}\Lambda_{m,k}\Gamma_{k,t}\right)\\ \; & +\frac{1}{T}\sum_{k_{1}=1}^{K}\sum_{k_{2}=1}^{K}\sum_{t=1}^{T-1}{\left(\Gamma_{k_{1},t+1}-P_{k_{1},k_{2}}\Gamma_{k_{2},t}\right)}^{2}, \end{array} \end{array} \end{array}$$
which is subject to the following constraints: (5)–(10). As previously mentioned, in Problem (12) ℓ(x,μ,σ) refers to the negative log likelihood: (13)$$\begin{array}{c} \begin{array}{c} \ell \left(x,\mu ,\sigma \right)=\frac{1}{2}\log\left(2\pi \sigma^{2}\right)+\frac{{\left(x-\mu \right)}^{2}}{2\sigma^{2}}, \end{array} \end{array}$$
and $P\in R^{K,K}$ is a matrix that influences the transition between different clusters. It is possible to use for *P* a stochastic transition matrix expressing the transition probability between each state and each other. For increased computational simplicity, we solve Problem (12) after applying Jensen’s inequality: (14)$$\begin{array}{c} \begin{array}{c} \begin{array}{cc} L_{\mathrm{eSPA}\text{-}\mathrm{Markov}} & =\frac{\varepsilon_{L}}{T}\sum_{d=1}^{D}\sum_{k=1}^{K}\sum_{t=1}^{T}W_{d}\Gamma_{k,t}\ell \left(X_{d,t},\mu_{d,k},\sigma_{d,k}\right)\\ \; & +\varepsilon_{E}\sum_{d=1}^{D}W_{d}\log\left(W_{d}\right)\\ \; & -\frac{\varepsilon_{CL}}{T}\sum_{m=1}^{M}\sum_{t=1}^{T}\Pi_{m,t}\sum_{k=1}^{K}\log\left(\Lambda_{m,k}\right)\Gamma_{k,t}\\ \; & +\frac{1}{T}\sum_{k_{1}=1}^{K}\sum_{k_{2}=1}^{K}\sum_{t=1}^{T-1}{\left(\Gamma_{k_{1},t+1}-P_{k_{1},k_{2}}\Gamma_{k_{2},t}\right)}^{2} \end{array} \end{array} \end{array}$$
which is subject to the following constraints: (5)–(10). We will now illustrate similarities and differences between this formulation and eSPA/FEMH1. The main framework is that of eSPA+, i.e., we can solve supervised learning problems, with the difference that we do not use matrix $S\in R^{D\times K}$ to store the values of the centroids for each cluster, which is instead replaced by two new terms: $\mu \in R^{D\times K}$ and $\sigma \in R^{D\times K}$. The first is the counterpart of *S* in eSPA, which contains the position of the centroids for each cluster, and the second is a matrix of variances for each cluster and each dimension. Note that we propose reconstructing the data using combination of univariate Gaussian distributions, as we sum the log likelihood in each dimension independently. This is a rather parsimonious assumption as the Gaussian distribution is the least informative distribution that can be used, following the maximum entropy principle. Having mean and variances, instead of minimizing the squared distance between each data point and the discretized representation as performed in eSPA+, we instead minimize the sum of the negative log likelihood of each data point given the mean and variance for each cluster, scaled by the affiliation probability and the feature probability vector *W*.

As with eSPA, we express the affiliation using the matrix Γ, but with the difference that each entry is allowed to be ∈[0,1] instead of being binary. This has the great advantage of allowing fuzzy clustering, despite coming with the cost of the absence of an analytical solution.

Another difference is the addition of a more general formulation of regularization through the ordering, which can be juxtaposed to the H1 term, which is used in FEMH1 to bias the solution towards a solution that is sufficiently “smooth” [34]. However, we introduce a transition penalization matrix $P\in R^{K\times K}$, which allows us to solve more general classes of problems. Indeed, in FEMH1, the assumption imposed is that the states are persistent, while the introduction of *P* allows different persistence levels. By choosing *P* to be an identity matrix of size *K*, we reduce this regularization term to that of FEMH1, which is what was used throughout this work.

Another difference is the placement of the coefficients that regulate the importance of each sub-problem, with the problem being a multicriteria optimization problem. In this work, the coefficient is in front of the likelihood term rather than in front of the H1 term, as the former can assume values from a very large interval. Thus, this change in coefficient grants increased numerical stability to the optimization of the entire problem.

### 2.2. Solutions of the Individual Steps in the Optimization Problem

In the following section, we will discuss the solution of each individual step, as well as the solution of the entire minimization problem, together with the computation of prediction from unseen data.

#### 2.2.1. Solution of μ and σ Steps

The procedure for finding the values of μ and σ that minimize the loss functional in Problem (14) can be treated together for simplicity. Both problems admit an analytical solution; namely, the optimal value for μ is found equivalently to that of *S* in eSPA, and the optimal value for σ is the standard deviation in each dimension, weighted by the cluster affiliations.

**Lemma 1.** 
*Given a value of Γ and the data matrix X, the values $\mu^{*}$ and $\sigma^{*}$ that minimize Problem (12) can be obtained as follows.*

*(a) The optimal value $\mu^{*}$, for a given cluster k and dimension d, can be found with:*

(15)
$$\mu_{d,k}^{*}=\frac{\sum_{t=1}^{T}\Gamma_{k,t}X_{d,t}}{\sum_{t=1}^{T}\Gamma_{k,t}}.$$


*(b) The optimal value for σ, for a given cluster k and dimension d, can be found with:*

(16)
$$\sigma_{d,k}^{*}=\frac{\sum_{t=1}^{T}\Gamma_{k,t}{\left(X_{d,t}-\mu_{d,k}\right)}^{2}}{\sum_{t=1}^{T}\Gamma_{k,t}}.$$



**Proof.** The proof of the previous two statements can be obtained by noting the following: (i) since the Gaussian distributions used for fitting are spherical, we can separate the problem into separate problems for each dimension; (ii) since each of the clusters are independent, the problem is separable in *K*. Then, we are left with problems that are a special case of fuzzy Gaussian mixture models, which can be solved through (15) and (16) [55].    □

#### 2.2.2. Solution of the Γ Step

The solution to the problem of finding a value of Γ that minimizes the loss functional is very similar to that of the equivalent Γ problem in FEMH1 [34], featuring some adjustments in the linear term only. Therefore, it is possible to use the efficient SPG-QP solver to calculate the updated value of Γ.

**Lemma 2.** 
*Given a dataset (X,Π) and fixed values of μ, σ, W, and *Λ*, the value $\Gamma^{*}$ that minimizes Problem (14) under the constraints (5) and (6) can be found numerically using the SPG-QP solver.*


**Proof.** We begin by constructing the column vector containing all values in Γ, which we denote as $\gamma \in R^{KT}$, (i.e., the vectorization of Γ). For constant values of μ,σ,W,Λ, we can rewrite problem (14) as:
(17)$$L_{\varepsilon }\left(\gamma \right):=b^{T}\gamma +\frac{1}{T}\gamma^{T}H\gamma .$$Matrix $H\in R^{KT\times KT}$ plays the same role as the homonymous matrix occurring when solving the FEMH1 problem, with the difference being that, in that case, it was a block diagonal matrix containing on its diagonal Laplace matrices. In our case, instead, the inclusion of matrix *P* in the problem has the effect of modifying matrix *H*, by potentially adding off-diagonal terms. It can be constructed as follows:
(18)$$H=I_{K}⊗\left[\begin{array}{ccccc} 0 & 0 & 0 & \ldots & 0\\ 0 & 1 & 0 & \ldots & 0\\ 0 & 0 & 1 & \ldots & 0\\ \vdots & \vdots & \vdots & \ddots & \vdots \\ 0 & 0 & 0 & \ldots & 1 \end{array}\right]+P^{T}P⊗\left[\begin{array}{ccccc} 1 & 0 & 0 & \ldots & 0\\ 0 & 1 & 0 & \ldots & 0\\ 0 & 0 & 1 & \ldots & 0\\ \vdots & \vdots & \vdots & \ddots & \vdots \\ 0 & 0 & 0 & \ldots & 0 \end{array}\right]+P⊗OD+P^{T}⊗OD^{T}$$
where $OD\in R^{T\times T}$ is defined as:
$$OD=\left[\begin{array}{ccccccc} 0 & -1 & 0 & \ldots & 0 & 0 & 0\\ 0 & 0 & -1 & \ldots & 0 & 0 & 0\\ 0 & 0 & 0 & \ldots & 0 & 0 & 0\\ \vdots & \vdots & \vdots & \ddots & \vdots & \vdots & \vdots \\ 0 & 0 & 0 & \ldots & 0 & -1 & 0\\ 0 & 0 & 0 & \ldots & 0 & 0 & -1\\ 0 & 0 & 0 & \ldots & 0 & 0 & 0 \end{array}\right].$$The linear term $b\in R^{KT}$ is a column vector containing the errors, including the modeling errors (negative log likelihoods) and the classification errors. It can be written as the vectorization of the following matrix:
(19)$$B_{k,t}:=-\left[\frac{\varepsilon_{L}}{T}\sum_{d=1}^{D}W_{d}\ell \left(X_{d,t},\mu_{d,k},\sigma_{d,k}\right)-\frac{\varepsilon_{CL}}{T}\sum_{m=1}^{M}\Pi_{m,t}\log\Lambda_{m,k}\right]$$Note that this problem is a constrained convex quadratic programming problem, and the constraints describe a closed convex set; thus, based on the extreme value theorem, it has a solution, which can be calculated using the SPG-QP algorithm, as shown previously [34].    □

#### 2.2.3. Solution of the Λ Step

The main difference between Problems (12) and (14) is the solution of the Λ step. Indeed, there exists no analytic solution for this step in the first case. Therefore, when solving Problem (12), it is necessary to use solvers based, e.g., on the interior point method. Instead, the solution of the Λ step in Problem (14) is equivalent to the solution of the Λ step in eSPA+. The optimal value $\Lambda^{*}$ that minimizes the loss functional (14) under the constraints (9) and (10) can be computed using:(20)$$\Lambda_{m,k}^{*}=\frac{{\hat{\Lambda }}_{m,k}}{\sum_{m=1}^{M}{\hat{\Lambda }}_{m,k}}.$$
where ${\hat{\Lambda }}_{m,k}=\sum_{t=1}^{T}\Pi_{m,t}\Gamma_{k,t}$.

#### 2.2.4. Solution of the *W* Step

The solution of the *W* problem, i.e., the value of $W^{*}$ that minimizes the loss functional (14) for a given collection of the remaining parameters can be obtained analytically in an equivalent way as the computation of the optimal value of *W* in eSPA+, and can be computed using the softmax function:(21)$$W^{*}=\frac{\exp-\frac{1}{T\varepsilon_{E}}b}{1_{D}^{T}\exp-\frac{1}{T\varepsilon_{E}}b}$$
where
(22)$$b_{d}=\sum_{t=1}^{T}\sum_{k=1}^{K}\Gamma_{k,t}\ell \left(X_{d,t},\mu_{d,k},\sigma_{d,k}\right).$$

#### 2.2.5. Solution to the Entire Problem

The solution to the entire optimization problem of minimizing (12) or (14) can be found through iterative subspace algorithm, and is illustrated in Algorithm 1. Label predictions for a new dataset $X_{test}$, given values of W,μ,σ,Λ and hyperparameters $\varepsilon_{E},\varepsilon_{L}$ can be obtained by solving for Γ once, as illustrated in Lemma 2, using a random feasible initial value, to obtain the affiliations to the clusters. Note that, in this case, not having a corresponding label probability matrix Π requires one to calculate the value of $\Gamma^{*}$ that minimizes Problem (14) under the constraints (5) and (6) while setting the value of $\varepsilon_{CL}$ to 0. Using the obtained Γ and the given matrix Λ, the label probabilities for each data point can be computed as:(23)$$\tilde{\Pi }=\Lambda \Gamma .$$
**Algorithm 1:** eSPA-Markov fitting algorithm
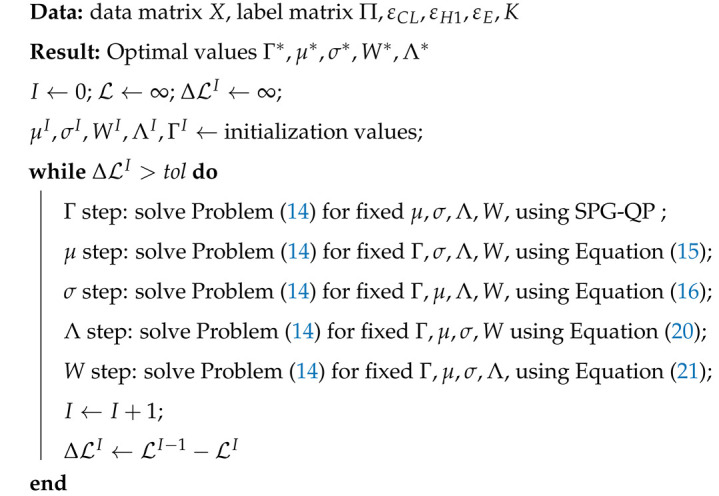


#### 2.2.6. Direct One-Shot Learning

The eSPA-Markov framework can be used also in a direct way to learn a model in a one-shot way (note: our definition of one-shot refers to learning the optimal parameters without iterative procedure, that can be applied to any other dataset originating from the same process, rather than the traditional definition employed in the field of large language model or computer vision), in the cases where it is possible to associate each centroid with a single label (and vice versa). Formal presentation of this procedure and the obtaining of assignments for any given test dataset is provided in the following theorems.

**Theorem 1.** 
*Let $X_{\mathrm{train}}\in R^{D\times T}$ be a given training dataset with T D-dimensional instances and $\Pi_{\mathrm{train}}\in {\left[0,1\right]}^{M\times T}$ be the corresponding label probability matrix, and let ${\left(\varepsilon_{E},\varepsilon_{L}\right)}_{n}$ be a fixed grid of hyperparameters with n grid points of possible combinations. If there exists a bijective mapping between the cluster affiliation *Γ* and the labels *Π*, all of the solutions that provide an infimum to the eSPA-Markov problem (11) can be computed with iteration cost scaling of O(DTMn), if the loss function L is a strictly convex function of θ.*


**Proof.** As there exists a bijective mapping between the centroids and the labels, the affiliation probabilities Γ and labeling probabilities Π are equal. That is, the probability of a data point being affiliated with a cluster is the same as the probability of that data point to possess a certain label. Given the equality
(24)$$\Gamma_{\mathrm{train}}=\Pi_{\mathrm{train}}$$
and the considered classification model Π=ΛΓ, we conclude that the optimal classification matrix $\Lambda^{*}$ is an identity matrix of size K=M. It is easy to check that this choice solves the Λ step in Algorithm 1, since the only term involving the variable Λ is the Kullback–Leibler divergence, and this term is equal to its minimal possible value, i.e., zero, for any choice of hyperparameter $\varepsilon_{CL}$. Therefore, for the objective function (11), we have
(25)$$\forall \;\theta ,\Lambda ,W:\;\;\;L\left(\Gamma_{\mathrm{train}},\theta ,\Lambda ,W\right)\geq L\left(\Gamma_{\mathrm{train}},\theta ,\Lambda^{*},W\right).$$Since Equation (24) provides the solution to the Γ problem for the training data, the solution with respect to θ is unique since L is strictly convex, according to the condition of the theorem.
(26)$$\forall \mu ,\sigma ,W:\;\;\;L\left(\Gamma_{\mathrm{train}},\theta ,\Lambda^{*},W\right)\geq L\left(\Gamma_{\mathrm{train}},\theta^{*},\Lambda^{*},W\right).$$The only remaining parameter to estimate is *W*, which depends on all previously computed (and fixed) parameters and on the hyperparameter $\varepsilon_{E}$. For any given $\varepsilon_{E}$ in the hyperparameter grid, it is therefore possible to find the optimal $W_{\varepsilon_{E}}^{*}$ using (21). Note that, for given $\varepsilon_{E}$ and its corresponding $W_{\varepsilon_{E}}^{*}$, we have
(27)$$\forall W:\;\;\;L_{\varepsilon_{E}}\left(\Gamma_{\mathrm{train}},\theta^{*},\Lambda^{*},W\right)\geq L_{\varepsilon_{E}}\left(\Gamma_{\mathrm{train}},\theta^{*},\Lambda^{*},W_{\varepsilon_{E}}^{*}\right).$$Notice that inequalities (25)–(27) hold for any $\varepsilon_{L}$ (which is not used so far since $\Gamma_{\mathrm{train}}$ is fixed), $\varepsilon_{CL}$ (which is not used in training since $\Lambda^{*}$ is identity and corresponding Kullback–Leibler term is zero), and each $\varepsilon_{E}$ (which defines an optimal $W_{\varepsilon_{E}}^{*}$).Afterwards, in order to obtain the optimal values of $\varepsilon_{L}$ and $\varepsilon_{E}$ that can be used on test data, we can directly compute a set of solutions for Problem (14) with Γ using SPG-QP for the given hyperparameter grid of $\varepsilon_{L}$ and $\varepsilon_{E}$ values. The result is a set of matrices $\Gamma_{\varepsilon_{E},\varepsilon_{L}}^{*}$, i.e., one for each combination of $\varepsilon_{E}$ and $\varepsilon_{L}$. Remember, when using a particular $\varepsilon_{E}$, we need to select the corresponding ${\hat{W}}^{\varepsilon_{E}}$. For fixed $\varepsilon_{L}$ and $\varepsilon_{E}$, we have
(28)$$\forall \Gamma :\;\;\;L_{\varepsilon_{E},\varepsilon_{L}}\left(\Gamma ,\theta^{*},\Lambda_{\mathrm{train}},W_{\varepsilon_{E}}^{*}\right)\geq L_{\varepsilon_{E},\varepsilon_{L}}\left(\Gamma_{\varepsilon_{E},\varepsilon_{L}}^{*},\theta^{*},\Lambda^{*},W_{\varepsilon_{E}}^{*}\right).$$The combination of (25)–(28) for $\Gamma =\Gamma_{\mathrm{train}}$ leads to optimality property (for any fixed selection of hyperparameters $\varepsilon_{E},\varepsilon_{L},\varepsilon_{CL}$)
(29)$$\forall \mu ,\sigma ,\Lambda ,W:\;\;\;L_{\varepsilon_{E},\varepsilon_{L},\varepsilon_{CL}}\left(\Gamma_{\mathrm{train}},\theta ,\Lambda ,W\right)\geq L_{\varepsilon_{E},\varepsilon_{L},\varepsilon_{CL}}\left(\Gamma_{\mathrm{train}},\theta^{*},\Lambda^{*},W_{\varepsilon_{E}}^{*}\right).$$At last, the optimal hyperparameters to be used on test data are the combination of $\varepsilon_{E}$ and $\varepsilon_{L}$ that minimizes the following error:
$$\left[\varepsilon_{E}^{*},\varepsilon_{L}^{*}\right]=\arg\min_{\varepsilon_{E},\varepsilon_{L}}{∥\Pi_{\mathrm{train}}-\Gamma_{\varepsilon_{E},\varepsilon_{L}}^{*}∥}_{2}$$At the end of the procedure, we obtain optimal values for the hyperparameters $\varepsilon_{E}^{*}$ and $\varepsilon_{L}^{*}$ as well as optimal model parameters $\theta^{*},\Lambda^{*},W_{\varepsilon_{E}^{*}}^{*}$, without any iterative algorithm. The computational cost of the procedure is of order O(DTMn), since we are solving *n* problems with cost O(DTM) exactly once. □

**Theorem 2.** 
*Given a single set of fixed parameters $\theta^{*},W^{*}$, hyperparameters $\varepsilon_{E}^{*},\varepsilon_{L}^{*}$ (e.g., determined applying Theorem 1 to a training dataset $X_{\mathrm{train}}$, $\Pi_{\mathrm{train}}$), and a test dataset $X_{\mathrm{test}}\in R^{D\times T}$, if $\Lambda^{*}$ is an identity matrix, and provided that the vectorization of the linear term in the QP problem is not in the image of the H matrix, there exists a unique solution of Problem (11) with respect to $\Gamma_{\mathrm{test}}$, which is computable with the overall complexity scaling $O(K^{p}T^{p}+KTD)$ and with iteration complexity scaling of O(KT).*


**Proof.** An assignment matrix $\Gamma_{\mathrm{test}}$ can be obtained as the minimizer of Problem (14) with respect to Γ, through the SPG-QP algorithm (as shown in Lemma 2). The solution to this problem would in the general case be unique if the *H* matrix (defined in Equation (18)) were to be symmetric positive definite. In our case, however, *H* is symmetric positive semidefinite.Since, per theorem conditions, the vectorization of the linear term is not in the image of the *H* matrix, we know that the solution is unique. This property can be easily proven by contradiction. Suppose that there exist two feasible different solutions $\gamma_{1*}\neq \gamma_{2*}$. Using the (exact) second-order Taylor expansion of the quadratic function (17), we have
(30)$$L_{\varepsilon }\left(\gamma_{1*}\right)=L_{\varepsilon }\left(\gamma_{2*}\right)+{\left(2H\gamma_{2*}-b\right)}^{T}\left(\gamma_{1*}-\gamma_{2*}\right)+{\left(\gamma_{1*}-\gamma_{2*}\right)}^{T}H\left(\gamma_{1*}-\gamma_{2*}\right).$$
Let us remember that the difference of solutions of convex quadratic programming problems is from the null space of the Hessian matrix (see, e.g., [56]); therefore, (30) can be simplified to
(31)$$\begin{array}{ccc} L_{\varepsilon }\left(\gamma_{1*}\right) & = & L_{\varepsilon }\left(\gamma_{2*}\right)+{\left(2H\gamma_{2*}-b\right)}^{T}\left(\gamma_{1*}-\gamma_{2*}\right)\\ \; & = & L_{\varepsilon }\left(\gamma_{2*}\right)-b^{T}\left(\gamma_{1*}-\gamma_{2*}\right). \end{array}$$
However, since *b* is not from the image of *H*, it can be written as b=bIm+bKer, where bIm∈ImH and $b_{\mathrm{Ker}}\in \mathrm{Ker}H$ with $b_{\mathrm{Ker}}\neq 0$. Then, the second term in (31) can be simplified to (using KerH⊥ImH):
bT(γ1∗−γ2∗)=bImT(γ1∗−γ2∗)+bKerT(γ1∗−γ2∗)=bKerT(γ1∗−γ2∗).
Notice that both of $b_{\mathrm{Ker}}$ and $\gamma_{1*}-\gamma_{2*}$ are non-zero; therefore, this term is non-zero. Consequently, by (31), we have $L_{\varepsilon }\left(\gamma_{1*}\right)\neq L_{\varepsilon }\left(\gamma_{2*}\right)$, which is a contradition with the optimality of $\gamma_{1*}$ and $\gamma_{2*}$ (both of them are supposed to be solution and therefore both of them have the same minimal objective function).Recall that the linear term *b*, in this case, consists of the vectorization of the error matrix:
(32)$$B_{k,t}=\sum_{d=1}^{D}W_{d}L\left(X_{d,t},\theta_{d,k^{\prime }}\right)$$
and that the null space of the *H* matrix, defined in (18), consists of constant functions. The condition is not particularly stringent, as it translates to a data matrix $X_{\mathrm{test}}$, which results in $B_{k,t}=B_{k^{\prime },t}$ for all $t,k,k^{\prime }$; in a case where this condition is not fulfilled, the solution exists but it is not unique.Concerning the computational cost, we need to consider two steps: (i) calculating the coefficients of the linear term and (ii) computing the gradient of the function in the iterative solver. The former is a fixed cost that occurs only once, as the coefficients for the linear term *b* are computed only at the beginning and requires a cost of O(KDT).Computing the gradient of the function (point ii) can be performed with O(KT) operations. Note, indeed, that if K<<T, which is often the case, then the resulting *H* matrix will be sparse, with at most $K^{2}$ off-diagonal elements, being a tridiagonal matrix if *P* is selected to be the identity matrix. Thus, the evaluation of the gradient reduces to multiplication between the sparse matrix and a vector of size KT.The global cost will be at maximum polynomial in K,T because, if H matrix is strictly convex, and the linear part is not in the kernel of H, then the QP problem can be solved in polynomial time [57]. Thus, the overall cost scaling is $O(K^{p}T^{p}+KTD)$ with p≥1, where the latter term is due to the computation of the Γ-independent coefficients *b* of the linear term in Equation (11). □

### 2.3. Experiments

All the experimental results were obtained using MATLAB (2024a), on a Dell PowerEdge R750 (Dell, Round Rock, TX, USA), with 2x Intel Xeon Platinum 8368 CPU (Intel, Santa Clara, CA, USA) (2.40 GHz), and 2048 GB of RAM memory.

### 2.4. Synthetic Data

The synthetic dataset is generating by simulating a Markov chain with two possible states and a transition matrix of the following form:$$P=\left[\begin{array}{cc} 1-\varepsilon & \varepsilon \\ \varepsilon & 1-\varepsilon \end{array}\right],$$
where ε was chosen to be a small number (0.01). This enforces persistence, i.e., each state to be more significantly likely to be followed by itself rather than the other one. In order to ensure the presence of both labels in each cross-validation split, rather than using a realization of a Markov process, we generated steps with stochastically determined length, centered at an interval that guarantees the visiting of both states in each partition of the dataset.

#### 2.4.1. Example 1

For each timestep, the system emits a measurement, sampled from a normal distribution with mean 0 and standard deviation being either $\sigma_{1}$ or $\sigma_{2}$, dependent on the state of the Markov chain. Namely, in the first state, the standard deviation was set to 1, and in the second state was varied, in order to obtain different $\sigma_{1}/\sigma_{2}$ ratios. The value of the ratio dictates the signal-to-noise ratio of the generated trace. A graphical representation of the projection on the first two dimensions of such a dataset is illustrated in the right panel of Figure 1B.

#### 2.4.2. Example 2

The measurement for each time step is collected from a bivariate normal distribution, which are generated using a covariance matrix of the form:$$C=\left[\begin{array}{cc} \varepsilon & 0\\ 0 & 1 \end{array}\right],$$
where ε controls the thickness of the distribution.

For the second state, the data was rotated with a rotation matrix with parameter α:$$R=\left[\begin{array}{cc} \cos(\alpha ) & -\sin(\alpha )\\ \sin(\alpha ) & \cos(\alpha ) \end{array}\right].$$
In this scenario, α controls the angle between the two distributions, such that π/2 refers to the case where the two distributions are superimposed with an angle of 90 degrees, i.e., they are orthogonal and simpler to separate, and in the π/16 case instead, the two distributions will be largely overlapping. A graphical representation of such a dataset is illustrated in Figure 1, in the right bottom panel.

#### 2.4.3. Synthetic RNA Sequencing Inspired Toy Example

This example is based by the task of denoising DNA or RNA sequencing traces obtained using Nanopore technology [58,59]. For simplicity, we generated an example reading using two bases (‘A’ and ‘B’), and with number of bases read at a given time (k-mer length) of 2. Each of their four possible combinations will correspond to a reading with a specific distribution of pore currents (shown in Figure 8, top row), and will persist in the pore for a dwell time that is selected from a combination-specific distribution. This example has a single dimension, and is used to illustrate the possibility to learn one-shot.

#### 2.4.4. Extra Dimensions

Multivariate datasets of this kind were obtain in identical manner, with the exception that a number of additional “uninformative” dimensions were appended, with the same number of sampled points. Those extra D−1 dimensions brought the dimensionality of the dataset to a total of *D*, and are sampled from the uniform distribution from the interval [0,1]. Thus, both examples are scalable in terms of number of instances, dimensions and “difficulty”, which for the first example means how large the noise is and for the second one instead how overlapping the two distributions are.

### 2.5. Preprocessing

#### Embedding

In order to provide time information to ML models that do not include order into account, we performed time embedding of the datasets with a variable window length [60]. The length of the window was selected independently for each of those models as an hyperparameter using cross-validation from the following values: 10, 50, and 100. This is a version of Takens’ delayed embedding reconstruction that operates on multivariate data in the following way: (33)$$\begin{array}{c} \begin{array}{c} \phi (x(:,t))=[x(:,t),x(:,t-1),\ldots ,x(:,t-l)], \end{array} \end{array}$$
where *l* represents the length of the embedding time window. As a result, from a single data point $x\left(t\right)\in R^{D}$, we obtain a new embedded representation of the data point $\phi \left(x\left(t\right)\right)\in R^{D\times (l+1)}$. Only data points where the entire window was available were considered.

### 2.6. Machine Learning Methods

For comparison, we used support vector machine (using the built-in MATLAB fitcsvm function) [61], random forest (using the built-in MATLAB treebagger function) [62], and deep learning. For deep learning, we used either a CNN network with a 1-D convolution layer, or a CNN-LSTM network the consisted of the same network with an addition of an LSTM layer and dropout [63]. For example, 1, we also included the same CNN-LSTM network which was provided with the wavelet transform of the input data [64].

#### Cross-Validation

In order to characterize the performance of each method on the problems at hand, we used cross-validation with triple splitting of the data. For every of the 20 cross-validation fold, we generated a dataset and split it in three equal parts, to be used as training, validation, and test. Note that the reported *T* in this work refers to the training data size, and not the total generated dataset size.

Hyperparameter selection was performed by exhaustively searching every possible hyperparameter combination for each method, with the grid reported in Table 1, and training them using the training dataset. Quality of the prediction was evaluated using the area under the receiver operating curve (AUC), a metric that describes the discriminatory skill of a classifier. AUC values close to 0.5 denote chance level predictions, while instead 1 indicates perfect classification. In contrast to other common metrics (for example, accuracy), it enables robust performance assessments in situations of (strongly) imbalanced data [41]. Then, the model with the best AUC when predicting the validation set was selected and its performance was evaluated on the test set. The training time refers to the training cost of the best model. The model labeled as “w-CNN-LSTM” refers to the same architecture and grid employed for the CNN-LSTM model, but with input provided as the wavelet transform of the data, using 12 voices per octave. eSPA-Markov was performed from 10 random initial points for each parameter combination. For the decision tree, the predictor selection “interaction-curvature” was only used for Example 1.

## 3. Results

We will now describe the results of benchmarking the eSPA-Markov algorithm in comparison to the methods introduced in the previous section. In Figure 2 and Figure 3, we present results obtained on the example number 1 (described in Section 2.4.1) for common machine learning and deep learning methods (respectively) and a fixed value of dimensionality (*D*) of 1. This condition therefore does not require the models to identify the relevant dimension within confounding ones, but reveals the ability to learn from the given time series. We trained all methods with datasets of increasing size in terms of available statistics, starting from an extremely limited amount of data (250 data points), to a moderate size (1000 data points) to a more conspicuous amount of training samples (2500 data points).

The goal of this comparison was not only to characterize the impact of limited amount of training data on the performance of the various algorithms, but also to investigate the effect of varying the signal-to-noise ratio of the data. Indeed, we repeated the comparisons for different values of σ-ratio, which is the ratio between the standard deviations of the distribution of the data in the two different states. An increase in the σ-ratio corresponds to decreasing levels of noise in the recording. This way, we can obtain discrimination curves for each method for the three conditions of data availability, which are plotted in the upper three panels. As expected, the performance for any method is reduced to random choice (AUC value close to 0.5) when the ratio is equal to 1 (i.e., there is no difference between the two states), as under those circumstances it is not possible to learn any distinction. It is possible to notice how, for each method, more training data leads to better performance in terms of the area under the ROC curve (AUC) on previously unseen test data. Note how the generalization ability of the investigated models, as illustrated with the difference in AUC between the training and testing sets, can increases with decreasing noise and data size. This is particularly noticeable for RF, while eSPA-Markov displays closer train and test scores. In the bottom three panels, instead, we report the computational cost for training the models, and illustrate the scaling of computational cost with respect to the level of noise and of the increase in the size of the data.

In Figure 4 and Figure 5, we report results obtained using datasets generated with the process described in example 1, the same as the previous figure, with the difference that in this case the number of dimensions used was 10. In order to successfully perform prediction of the labels associated with the data points, it is therefore necessary to identify the single relevant dimension in which there is a difference between the two classes while being provided with nine additional uninformative dimensions. Results are plotted in the same way as Figure 2, including comparison of the AUC on the test dataset in the top row, the generalization ability in the middle row and the computational cost in the bottom row by varying the value of σ-ratio at three different levels of data size.

In Figure 6 and Figure 7, we report the results on example 2 (described in Section 2.4.2), keeping a fixed amount of available statistics for training (1000), but varying the dimensionality of the dataset. The leftmost plots represent datasets with dimension (D) equal to 5, the middle ones instead have dimension equal to 100 and on the right the dimensionality is instead of dimension 500. This comparison captures the performance and the computational cost of fitting the different algorithms for data that is increasingly “small” (using the definition provided in Section 1.2.1). For this example, we examined the performance by varying the angle between the two distributions (α). Note that in comparison, smaller values of α (on the right) represent a more challenging situation.

Figure 8 depicts the outcome of applying the direct one-shot learning methodology, on a toy example inspired by Oxford Nanopore sequencing.

This example is of particular relevance, as this type of data represents a paradigm shift in sequencing technology, enabling the reading of long sequences in the form of noisy recording of current [65,66]. Briefly, this molecular biology technique allows the sequencing of nucleic acid polymers to identify the order of the bases. To do so, individual molecules are moved across an impermeable membrane through a pore, while the current across the pore can be read. Within the pore, at any given time of the passage of the molecule, it is possible to find a certain (fixed) number of bases, which is called the length of the k-mer. For our toy example, we simulated an experiment as described above, with two possible bases (“A” and “B”) and a k-mer length of 2.

Each combination of bases will correspond to a unique distribution of currents that can be read by the device, which we simulated by introducing for each combination a specific mean and standard deviation. Moreover, as the passage though the pore is slower than the sampling rate of the instrument, each position along the macromolecule will correspond to several readings. It is also to be noted that each possible combination tends to exhibit a specific dwell time, i.e., the typical length of the sampling is dependent on the combination. This can be seen in Figure 8, as, e.g., the combination “BA” (colored in green) typically has less persistent (shorter) reading times compared the combination “BB” (in blue).

As each pore in the machine is reading an independent signal, the data are generally a collection of single-dimensional vectors rather than a multivariate unified dataset. Therefore, in this case, there is no need to employ *W*, as it can be considered to be equal to 1 for the only dimension present.

By application of the direct one-shot learning methodology, we obtain near-perfect reconstruction, in virtue of being able to select the optimal value for the hyperparameter (in this case, only $\varepsilon_{L}$, since the data has one dimension). Application of this model to test data, which were generated starting from a different sequence, also provides near-perfect reconstruction of the labels for each time instance (right panels).

## 4. Discussion

In this work, we formulated and analyzed the eSPA-Markov algorithm, which expands the family of Entropic AI methodologies to perform learning on ordered data, by addition of a regularization term that biases the solution to have “smooth” transitions between the clusters—where “smoothness” is understood in the sense of the closeness to the latent trajectories of an underlying Markov process. The Markovian smoothness in the learning procedure is enforced through an additional regularization term, which allows applying different forms of priors on the latent Markovian process, given in terms of the Markovian transition matrix *P*. For example, by selecting the prior matrix *P* as an identity matrix, we can enforce persistence/metastability of the latent process.

We illustrated how using the one-dimensional negative log-likelihood as the loss function allows us to distinguish between regimes with overlapping means and differences in variance. We simultaneously solve the problems of learning the segmentation patterns as well as the entropy-optimal feature space discretizations and Bayesian classification rules, via iterative monotonic minimization of the constrained functional (12). In Theorems 1 and 2, we provided estimates of the computational cost scaling and conditions for existence and uniqueness of the solutions of the resulting problem.

The performance of this algorithm on toy examples with time evolution was compared to state-of-the-art machine learning, deep learning, and signal processing tools, including neural networks that feature convolutional layers and LSTM layers, as well as with the preprocessing using wavelet transform. With the provided examples, we highlight how the proposed eSPA-Markov approach not only allows classification of data in spite of elevated noise but it does so with a competitive computational performance, and on high-dimensional data.

In order to obtain a fair and thorough comparison between the different methods, we first focused on univariate data, in which successful prediction does not require the identification of the relevant dimensions. Please note that the dimensionality reduction step is explicitly included in our methodology but is not explicitly available in the other considered methods—so, this should give an advantage to the selected common tools in one-dimensional applications. We compared the performances while varying the signal-to-noise level and the number of available data points for training (T). As can be seen from Figure 2 and Figure 3, the performance of common tools deteriorates quickly when approaching the small noisy data scenario, whereas eSPA-Markov outperforms the competition in this situation. Limited training data represent a challenging condition, as can be seen when the available data points for training are 250 (see Figure 2). eSPA-Markov consistently obtained higher scores than every other method, as can be seen particularly for σ-ratios of 1.5 with *T* being 1000 or 2500, where the proposed method is the only one out of the compared methods that could consistently reach an AUC close to 1. In terms of computational cost, one observes (i) an increase in cost for all the considered methods for growing data statistics, and (ii) that the cost scaling of eSPA-Markov is comparable to that of SVM, which is traditionally appreciated for its advantageous cost scaling, in situations with restricted sample sizes and elevated dimensionality.

As shown in Figure 4 and Figure 5, the performance of common tools drops drastically with the addition of a modest number of non-informative dimensions. Indeed, despite the relatively small difference from the previous comparison, adding non-informative data dimensions is sufficient to impair the performance of the considered deep learning methods that fail to produce good results on the testing set. This is a manifestation of the well known problem of overfitting, as the networks achieve perfect prediction on the training split, but are not able to generalize their performance when using unseen test data. This issue is due to the fact that increasing the number of dimensions requires the use of more data to form accurate decision boundaries [22]. In fact, for all of the methods, larger training datasets allowed better performance on the test data; for example, CNN-LSTM was only able to provide valuable predictions for the larger dataset sizes we evaluated. Providing the wavelet-transformed data to the CNN-LSTM network (i.e., in the “w-CNN-LSTM” model) could partially mitigate this issue and allow a better performance on small datasets, in a comparable way to random forest models. However, the performance of eSPA-Markov was shown to always be superior to the other models taken into account, and it was able to consistently obtain better predictions for the whole range of training data sizes. Moreover, the computational cost of training eSPA-Markov was one order of magnitude lower than that of RF and two orders of magnitude lower than that of w-CNN-LSTM in the low-data-availability condition. Summarizing, eSPA-Markov displayed a competitive performance, outperforming considered common tools in terms of quality of prediction, training cost, and scalability.

Figure 6 and Figure 7 further illustrated the performance in a situation of increasingly “small” data. It was possible to observe how the performance of deep learning techniques as well as of SVM quickly degraded with the increasing number of dimensions. It is important to note that the cost of SVM being very low on the datasets with higher dimensionality was negated by the poor predictive performance on these datasets. As for the previous example, random forest was the only method besides of the eSPA-Markov that could provide good predictive performance, but with more than one order of magnitude larger computational costs across all examined feature sizes. These results do not imply that larger neuronal networks (or networks with different architectures) could not perform better than the ones selected here, but finding an optimal network architecture and hyperparameterization is an NP-hard task. In contrast, finding the optimal values for four gridded hyperparameters (namely, $K,\varepsilon_{CL},\varepsilon_{E}$ and $\varepsilon_{L}$) scales polynomially (as the polynomial of degree 4, when checking all possible hyperparameter combinations), with the cost of every hyperparameter combination scaling linearly (according to the Theorem 1).

We also demonstrated the possibility of using the proposed framework in a direct one-shot way, in the case when there exists a bijective mapping between the label probabilities and the cluster affiliations. We showcased the efficacy of this task on a toy model, based on a real-life application in molecular biology/bioinformatics—the denoising and classification of sequencing data obtained with the Oxford Nanopore technology. Scaling this approach up to real datasets of this type, which would involve a much larger number of possible latent states *K* (due to the higher number of bases, also considering epigenetics modifications, and k-mer length) represents an interesting future application of the presented methodology. Another potential use case is that of noisy data series having different notions of ordering—not necessarily being the ordering in time. For instance, in the field of molecular biology, measurements from cells could be ordered along the developmental trajectory of their differentiation, and if the samples cannot be measured with an explicit ordering, an approximation can be inferred from the data by using the permutation finding the smoothest reordering [67]. Additionally, any arbitrary graph could be used to formulate the respective Markovian prior *P*—imposing a notion of “neighborhood” and “smoothness” that is relevant for the problem at hand. As an example, eSPA-Markov could be used to obtain smooth reconstructions simultaneously in time and space, with application to images and videos. Thus, this technique can potentially match for supervised classification of the results obtained by its unsupervised variant, which was shown to outperform state-of-the-art methods in denoising extremely low radiation and high-noise computer tomography data [29].

## Figures and Tables

**Figure 1 entropy-26-00553-f001:**
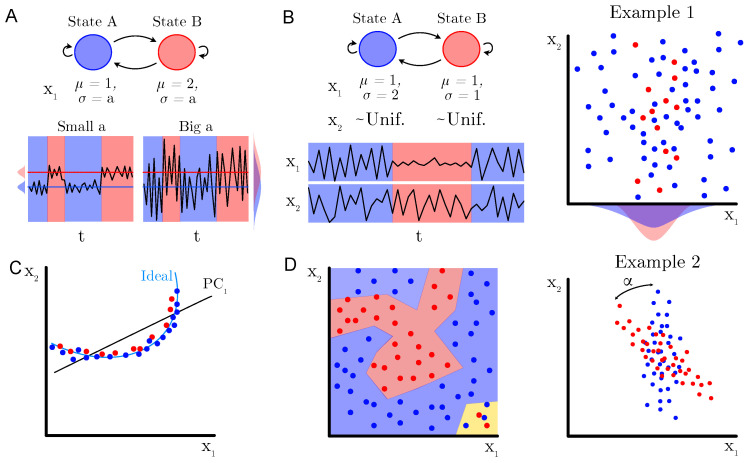
Graphical representation of the examples mentioned in the introduction for the denoising problem (**A**,**B**), dimensionality reduction (**C**), and nonlinear decomposition (**D**). On the right panel, one can see an illustration of the toy examples.

**Figure 2 entropy-26-00553-f002:**
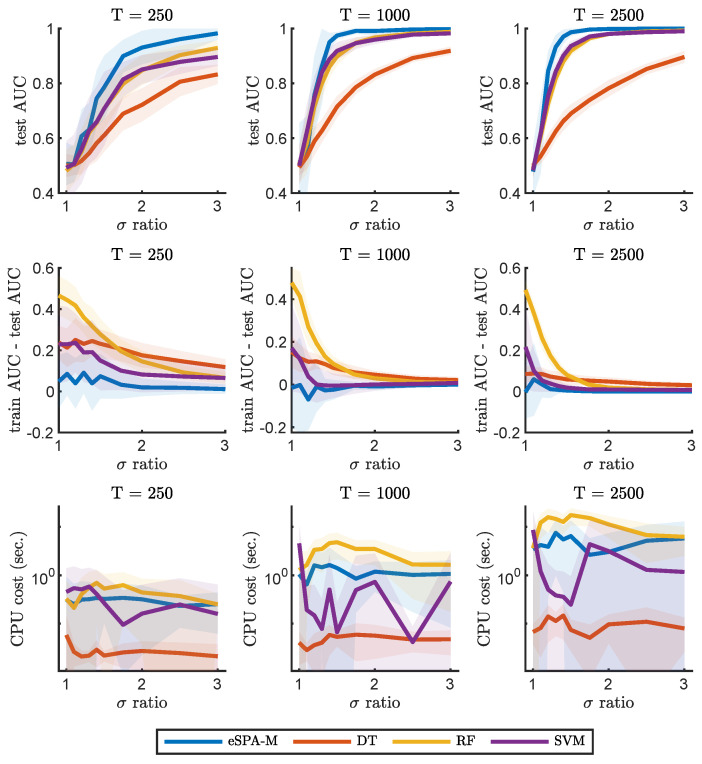
Comparison of eSPA-Markov (in blue) versus machine learning methods (decision trees, random forests, and support vector machines) on example 1 with D = 1. Upper row—performance (AUC on test data); middle row—difference in performance between the training set and the test set; lower row—computational time for three different training data sizes (indicated above). Results are presented as mean ± standard deviation (SD).

**Figure 3 entropy-26-00553-f003:**
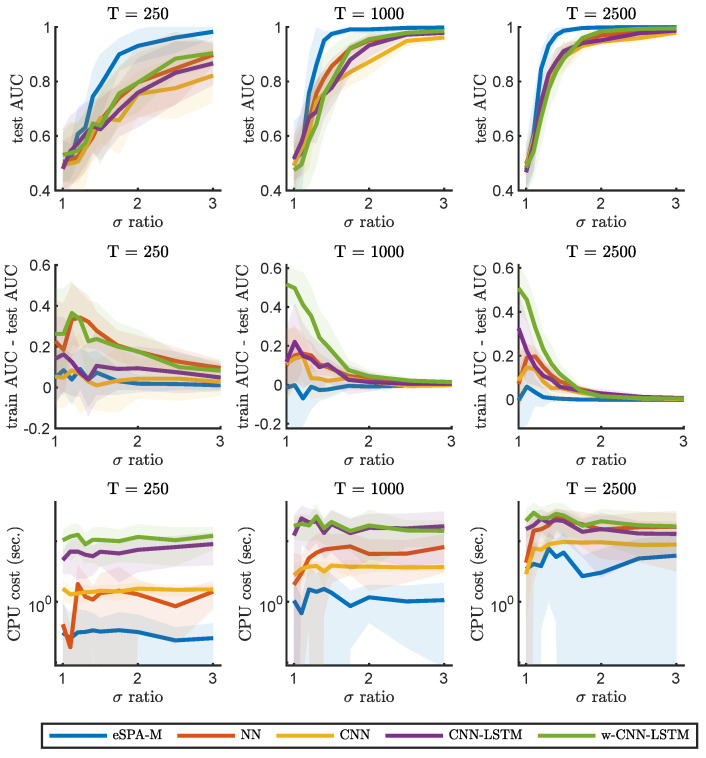
Comparison of eSPA-Markov (in blue) versus neural networks and deep learning methods (convolutional and LSTM networks) on example 1 with D = 1. Upper row—performance (AUC on test data); middle row—difference in performance between the training set and the test set; lower row—computational time, for three different training data size (indicated above). Results are presented as mean ± standard deviation (SD).

**Figure 4 entropy-26-00553-f004:**
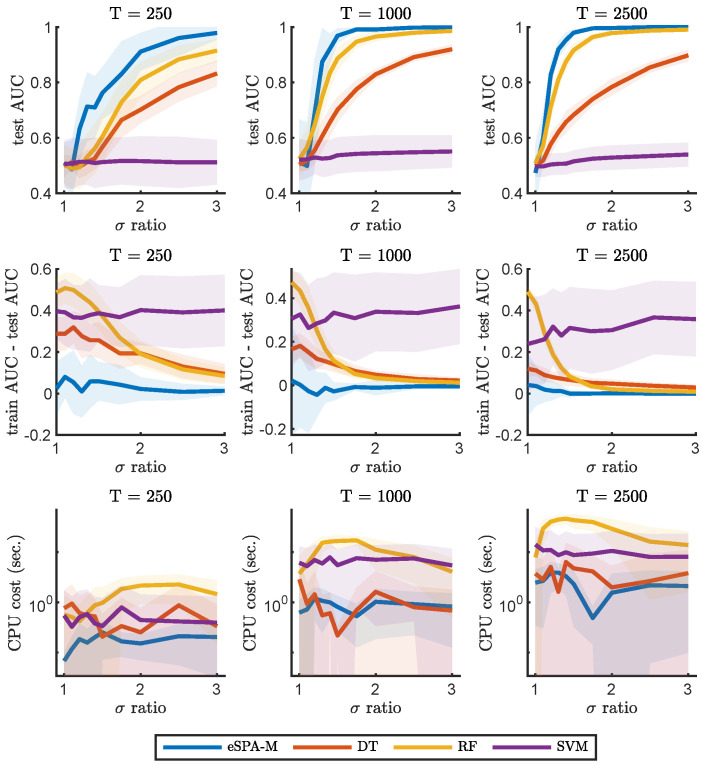
Comparison of eSPA-Markov (in blue) versus machine learning methods (decision trees, random forests, and support vector machines) on example 1 with D = 10. Upper row—performance (AUC on test data); middle row—difference in performance between the training set and the test set; lower row—computational time, for three different training data size (indicated above). Results are presented as mean ± SD.

**Figure 5 entropy-26-00553-f005:**
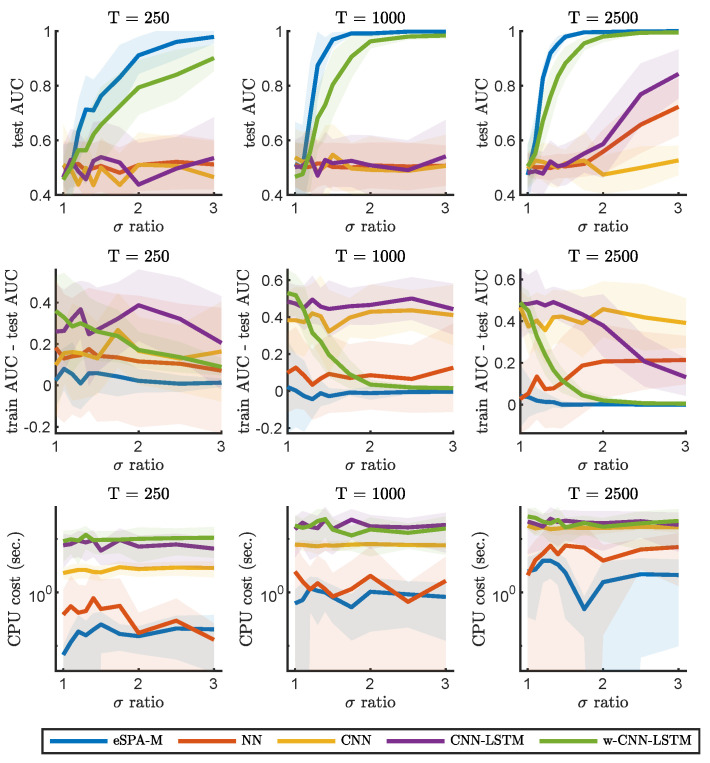
Comparison of eSPA-Markov (in blue) versus neural networks and deep learning methods (convolutional and LSTM networks) on example 1 with D = 10. Upper row—performance (AUC on test data); middle row—difference in performance between the training set and the test set; lower row—computational time, for three different training data size (indicated above). Results are presented as mean ± SD.

**Figure 6 entropy-26-00553-f006:**
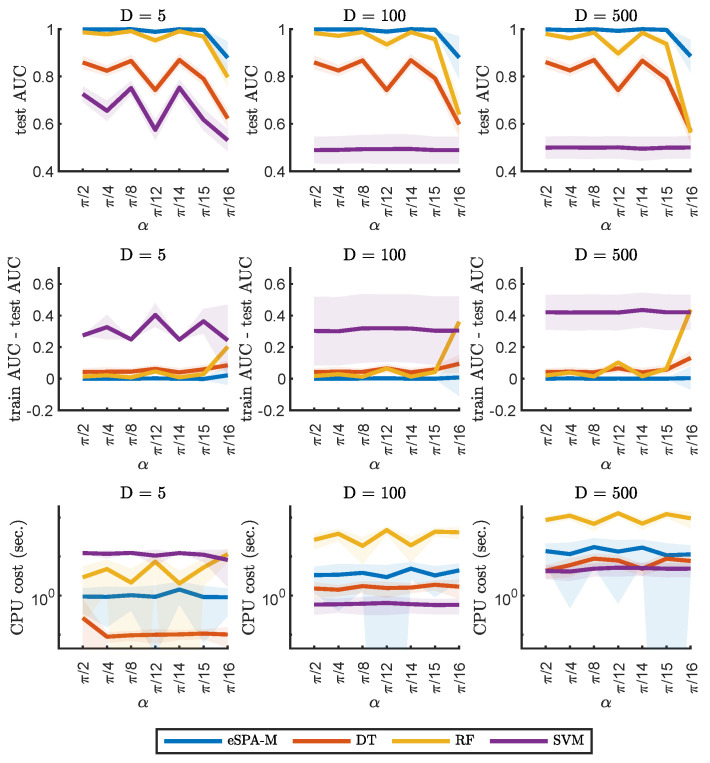
Comparison of eSPA-Markov (in blue) versus machine learning methods (decision trees, random forests, and support vector machines) on example 2. Upper row—performance (AUC on test data); middle row—difference in performance between the training set and the test set; lower row—computational time, for three different sizes of dimensionality of the dataset (indicated above). Results are presented as mean ± SD.

**Figure 7 entropy-26-00553-f007:**
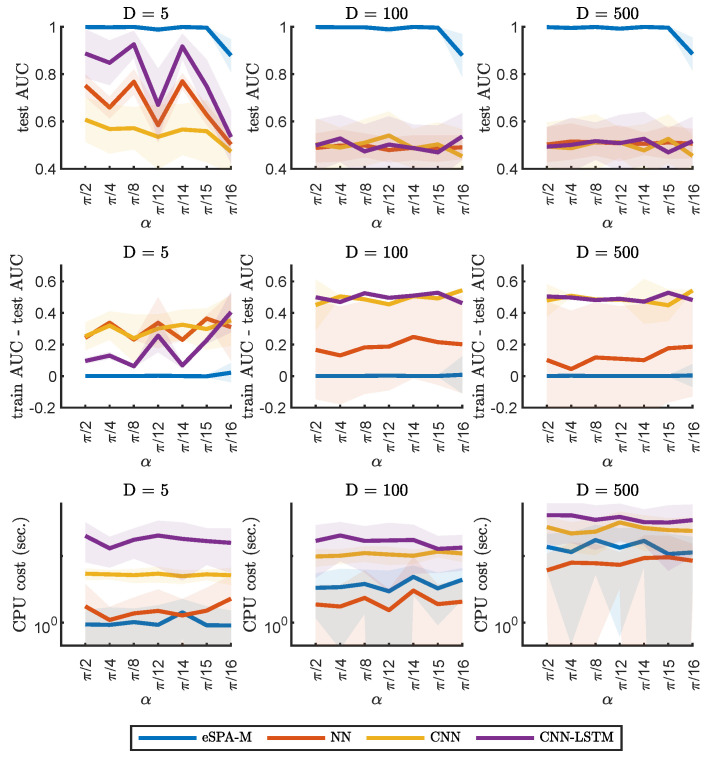
Comparison of eSPA-Markov (in blue) versus neural networks and deep learning methods (convolutional and LSTM networks) on example 2. Upper row—performance (AUC on test data); middle row—difference in performance between the training set and the test set; lower row—computational time, for three different sizes of dimensionality of the dataset (indicated above). Results are presented as mean ± SD.

**Figure 8 entropy-26-00553-f008:**
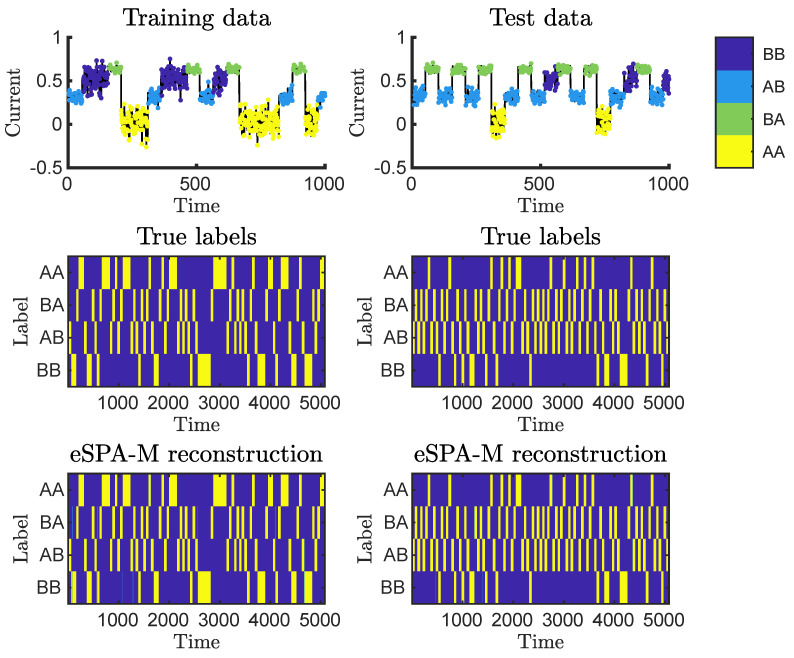
Synthetic RNA sequencing example. The panels on the left refer to training data and those on the right refers to previously unseen test data. The top row depicts the raw current, color-coded to represent the combination of two letter for a given time point. Middle and bottom panels represent, respectively, the true labels and the reconstruction obtained with the direct one-shot learning procedure.

**Table 1 entropy-26-00553-t001:** Hyperparameter grid.

Hyperparameter	Values
eSPA-Markov	
$\varepsilon_{CL}$	1 × $10^{-4}$, 1 × $10^{-3}$, 1 × $10^{-2}$
$\varepsilon_{L}$	1 × $10^{-5}$, 1 × $10^{-4}$, 1 × $10^{-3}$, 1 × $10^{-2}$, 1 × $10^{-1}$, 1, 2, 5, 10
$\varepsilon_{E}$	1 × $10^{-4}$, 1 × $10^{-3}$, 1 × $10^{-2}$, 1 × $10^{-1}$
K	2, 3, 5
Decision Tree (DT)	
Embedding window size	10, 20
Minimal leaf size	5, 10, 20, 30
Maximal number of splits	5, 10, 20
Predictor selection	‘allsplits’, ‘curvature’, ‘interaction-curvature’
Support Vector Machine (SVM)	
Embedding window size	10, 20
Kernel scale	$10^{-8},10^{-7},10^{-6},10^{-5},10^{-4},10^{-3},10^{-2},1,10^{2},10^{4}$
Box Contraint	$10^{-5},10^{-4},10^{-3},10^{-2},10^{-1},1,2,4,5,6,8,10$
Random Forest (RF)	
Embedding window size	10, 20
Minimal leaf size	5, 10, 20, 30
Number of trees	25, 50, 75, 100, 250, 500, 1000
Splitting criterion	gdi, deviance
Support Vector Machine (SVM)	
Embedding window size	10, 20
Kernel scale	$10^{-8},10^{-7},10^{-6},10^{-5},10^{-4},10^{-3},10^{-2},1,10^{2},10^{4}$
Box Contraint	$10^{-5},10^{-4},10^{-3},10^{-2},10^{-1},1,2,4,5,6,8,10$
Neural Network (NN)	
Embedding window size	10, 20
Activation function	relu, sigmoid
Layer size	1, 10, 25, 50, 100, [10, 10], [25, 25], [50, 50], [100, 100]
Lambda	0, 1 × $10^{-3}$, 1 × $10^{-2}$, 1 × $10^{-1}$
Convolutional Neural Network (CNN)	
Embedding window size	10, 20
Learning rate	0.001
Patience	10
Filter size	3, 5, 10
Number of filters	1, 3, 5, 10
Convolutional Neural Network + Long Short Term Memory (CNN-LSTM)	
Embedding window size	10, 50
Learning rate	0.001
Patience	10
Filter size	3, 5, 10
Number of filters	1, 3, 5, 10
LSTM layer size	100, 250, 500
Dropout	0, 0.1

## Data Availability

The code used in this work can be shared upon request to the authors.

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
