# Peer review of "On Entropic Learning from Noisy Time Series in the Small Data Regime"

_entropy, 2024, doi:10.3390/e26070553_

Round 1
Reviewer 1 Report
Comments and Suggestions for Authors
The authors develop the method for short data time series models. In fact, much effort have been made in "big data" analysis and indeed most of the AI models require significant amount of data for model training, particularly that the existing data have to be divided into three chunks: learning, test and verification. On the other hand there are situation where the huge data sets are not available or very difficult to collect (due to the instabilities of the system, the size of data sets etc.). In scientific literature there exists the problem of estimation on "data sample" which is related to the problem of small size of data sample. Therefore this subject is interesting for many scholars.
The presented paper provide modification of ML algorithm making it suitable for usage on relatively short time series. Very important advantage of the paper is that the authors not only define the model and verify it on some chosen examples but also discuss it rigorously.
Considering the improvements/corrections of the paper - there are only a few misprints which could be easily corrected on the galley prove stage:
1. eq. 20 \hat{\Lambda} is not defined
2. the same eq. 21 b'
3. In the introduction. AR and MA models are standard time series models and certainly are not ML models. One can refer to the time series models, but it is not necessary.
The strength of the paper is the quality of the presentation. The model is discussed also with mathematical details, which is not often in machine learning papers. However, simulation results are not spectacular.
In my opinion, the results obtained by the authors are suitable for publication.
Author Response
Reviewer # 1
The authors develop the method for short data time series models. In fact, much effort have been made in "big data" analysis and indeed most of the AI models require significant amount of data for model training, particularly that the existing data have to be divided into three chunks: learning, test and verification. On the other hand there are situation where the huge data sets are not available or very difficult to collect (due to the instabilities of the system, the size of data sets etc.). In scientific literature there exists the problem of estimation on "data sample" which is related to the problem of small size of data sample. Therefore this subject is interesting for many scholars.
The presented paper provide modification of ML algorithm making it suitable for usage on relatively short time series. Very important advantage of the paper is that the authors not only define the model and verify it on some chosen examples but also discuss it rigorously.
Considering the improvements/corrections of the paper - there are only a few misprints which could be easily corrected on the galley prove stage:
1. eq. 20 \hat{\Lambda} is not defined
2. the same eq. 21 b'
The two points have been corrected. Many thanks for pointing them out.
In the introduction. AR and MA models are standard time series models and certainly are not ML models. One can refer to the time series models, but it is not necessary.
We have updated the sentences in the introduction that refers to AR and MA models, and we explicitly refer to them now as time series models.
The strength of the paper is the quality of the presentation. The model is discussed also with mathematical details, which is not often in machine learning papers. However, simulation results are not spectacular.
In my opinion, the results obtained by the authors are suitable for publication.
Reviewer 2 Report
Comments and Suggestions for Authors
The following is my review of the paper entitled “On entropic learning from noisy time series in the small data regime” submitted to the MDPI journal “Entropy”.
The paper’s topic is interesting, and it perfectly fits the journal’s scope.
Here are my comments:
1. Why do the authors put the "time" word between parentheses? It is quite obvious that the data they are talking about is ordered according to time. As a matter of fact, I do not understand why the authors do the same for several terms. I suggest removing this. It might be confusing.
2. Why do the authors not use the term "time series" to refer their data? As far as I see there is no difference between the two terms. It should be clarified.
3. This phrase is not grammatically correct: "Let us imagine to identify" line 55 page 2.
4. There are five pages of introduction that though interesting, I believe they might be summarised. The important point starts from page 5, section 1.2.3.
5. Maybe I missed something but although the authors create synthetic data, they do not mention how many data was created. How may samples per dataset? (They mentioned this at the end, "We trained all methods with datasets of increasing size in terms of available statistics, 517 starting from an extremely limited amount of data (250 data points), to a moderate size 518 (1000 data points) to a more conspicuous amount of training samples (2500 data points). ". It should be introduced earlier.
6. Also, I did not understand why half of the paper is talking about clustering methods and then they implemented classification models eventually.
7. It should be properly introduced how the experiments are going to be evaluated. There is a mixture of concepts in the results section that must be introduced beforehand (AUC).
8. The authors emphasise a lot the time-thing feature of the data but they are not dealing with this at all. After all they are only classifying, and it is not clear the goal of the classification for me. What is the target?
9. Why SVM model is skipped in some experiments?
10. This claim is not completely true "SVM, which is traditionally appreciated for its low cost." If the model does not find the hyperplane it may take hours of iterations.
11. This other claim is obvious "Deep Learning methods that fail to produce good results on the testing set" as the same authors say somewhere in the text, those methods needs more data to be useful. Actually, I would say the comparison is a bit unfair. If the authors' method operates with few data, then they ought to compare with simpler models, like simple trees, perceptron, maybe MLP and so on, as well.
12. Finally, the authors at the end, they also say that "this technique can potentially match for supervised classification of the results obtained by its unsupervised variant". I do not understand if these are two different problems, with different approaches to address the problem to solve (supervised and unsupervised), why do the authors speak of them interchangeably? It should be clarified, otherwise it seems to be a misunderstanding of concepts.
13. This paper addresses the challenge of managing perishable vegetables in supermarkets. Due to their short shelf life, reselling them is difficult, making accurate demand forecasting critical for maximizing profits. The authors propose a new approach for optimizing pricing and replenishment decisions in vegetable departments using LSTM and genetic algorithms.
14. The authors mentioned at the beginning of paper that they studied the scalability of their method. I do not consider the experiments done analyse properly this feature.
Comments on the Quality of English LanguageI detected some minor typos.
Also the use of parentheses between terms is excesive from my point of view (see comments).
Author Response
Reviewer 2
The following is my review of the paper entitled “On entropic learning from noisy time series in the small data regime” submitted to the MDPI journal “Entropy”.
The paper’s topic is interesting, and it perfectly fits the journal’s scope.
Here are my comments:
- Why do the authors put the "time" word between parentheses? It is quite obvious that the data they are talking about is ordered according to time. As a matter of fact, I do not understand why the authors do the same for several terms. I suggest removing this. It might be confusing.
- Why do the authors not use the term "time series" to refer their data? As far as I see there is no difference between the two terms. It should be clarified.
Thanks, we fixed it. We would like to point out that the meaning behind the parenthesis was not to suggest the reader that the ordering is in time, but rather to highlight how time is only one of the possible data orderings that can be used. Even if in this work we apply the described model to only time ordered data, it can potentially be applied to data that is ordered in other ways. Pseudotime- and developmental time-ordered data from the field of biology or data ordered on any arbitrary graph are concrete examples of what we mean. We do, however, understand how it could sound confusing, so we modified the manuscript to improve clarity.
- This phrase is not grammatically correct: "Let us imagine to identify" line 55 page 2.
The sentence was corrected.
- There are five pages of introduction that though interesting, I believe they might be summarised. The important point starts from page 5, section 1.2.3.
Thanks for the suggestion. We considered shortening the introduction; however, we were unable to agree on which parts it should be shortened. We still think that the current version provides helpful information for the readers, and that in its current formulation it allows clarifying the link between clustering and the regularization used in this study (one of the points raised in another reviewer’s comment).
- Maybe I missed something but although the authors create synthetic data, they do not mention how many data was created. How may samples per dataset? (They mentioned this at the end, "We trained all methods with datasets of increasing size in terms of available statistics, 517 starting from an extremely limited amount of data (250 data points), to a moderate size 518 (1000 data points) to a more conspicuous amount of training samples (2500 data points). ". It should be introduced earlier.
- The authors mentioned at the beginning of paper that they studied the scalability of their method. I do not consider the experiments done analyse properly this feature.
The answers to these two points are joined in one. The number of generated data points is mentioned in the results section because one of the aims of our experiments was to investigate the scaling of the methods, which was provided both in the theorems and lemmas – as well as in the figures (lower panels of figures 2,3 and 4). Therefore, for each crossvalidation and combination of D and T a dataset was generated with according dimensions. That is why we indicate the (changing) sizes of datasets in terms of features and samples in each plot and before discussing the relative results rather than in the methods section.
- Also, I did not understand why half of the paper is talking about clustering methods and then they implemented classification models eventually.
- The authors emphasise a lot the time-thing feature of the data but they are not dealing with this at all. After all they are only classifying, and it is not clear the goal of the classification for me. What is the target?
- Finally, the authors at the end, they also say that "this technique can potentially match for supervised classification of the results obtained by its unsupervised variant". I do not understand if these are two different problems, with different approaches to address the problem to solve (supervised and unsupervised), why do the authors speak of them interchangeably? It should be clarified, otherwise it seems to be a misunderstanding of concepts.
As for the previous answer, we address these three points together. Thank you for raising these issues, we modified the text accordingly (please see page 6). The method we introduce, as well as those that it builds upon, is derived from clustering methods. Therefore, we believe they are a natural place for beginning our explanation. As described in the introduction, the proposed technique is derived from previously published methodologies, namely eSPA and FEMH1. The latter, in particular, performs unsupervised denoising of time series using the H1 semi-norm as regularization term, exhibiting excellent performance (reported in previous literature). As explained in the introduction, this approach can be seen as a clustering algorithm. However, the FEMH1 formulation did not allow the solution of supervised problems (and it did not allow dimensionality reduction). eSPA+ also dwells on unsupervised clustering ideas, and can be applied to supervised learning through the addition of a regularization term based on the KL distance between the labels and the label probabilities returned by the model.
The consideration of the time structure of the data (which we assume is what the reviewer is referring to) is a crucial feature for the proposed method (as it was for FEMH1). Indeed, classification is performed leveraging the cluster affiliation vector of each data point.
The supervised problem and the unsupervised one are different problems, but mathematically, they are not so distant. For example, in eSPA and eSPA-Markov, by setting the value of $\varepsilon_{CL}$ to 0, we obtain a method that solves the unsupervised problem. Moreover, if the distance metric used in the discretization part is the Euclidean distance, and the other regularization coefficients are set to 0, eSPA becomes equivalent to the widely-used k-means clustering.
- It should be properly introduced how the experiments are going to be evaluated. There is a mixture of concepts in the results section that must be introduced beforehand (AUC).
We introduced a brief explanation of the AUC metric in the methods section.
- Why SVM model is skipped in some experiments?
- This claim is not completely true "SVM, which is traditionally appreciated for its low cost." If the model does not find the hyperplane it may take hours of iterations.
SVM was actually not skipped in any of the presented experiments. Concerning the sentence quoted in point 10, the computational complexity of SVM is between O(D \times T^2) and O(D \times T^3). Thus, SVM can have poor performance in case of large number of samples. The sentence was updated accordingly. SVM dwells on solving a quadratic programming problem, and therefore the convergence and the number of iterations needed can be described by analytical conditions [1]. In a nutshell, SVM will always converge in polynomial time if the underlying QP problem is strictly convex and the set of constraints is not in the kernel of the Hessian matrix. We mentioned these results in the proof of Theorem 4.
Hush, D., & Scovel, C. (2003). Polynomial-time decomposition algorithms for support vector machines. Machine Learning, 51, 51-71.
- This other claim is obvious "Deep Learning methods that fail to produce good results on the testing set" as the same authors say somewhere in the text, those methods needs more data to be useful. Actually, I would say the comparison is a bit unfair. If the authors' method operates with few data, then they ought to compare with simpler models, like simple trees, perceptron, maybe MLP and so on, as well.
“Small data”, as explained in the paper, is traditionally considered difficult to learn from. Indeed, as shown in previous work, Deep Learning methods can empirically display an overfitting barrier, which means that there exists a minimum amount of training data required to learn a specific task for a given number of dimensions.
Our investigation has the main goal of formulating methods that “push the limit” of what is possible to learn by successfully operating beyond the Deep Learning overfitting barrier. As such, it is of critical importance to (i) show where DL methods start to overfit and (ii) hope to improve the performance. We would like to stress that the aim of this work is simply to state how small data can be successfully analysed, which requires finding the overfitting limit for DL methods, and employing several methods under those conditions.
- This paper addresses the challenge of managing perishable vegetables in supermarkets. Due to their short shelf life, reselling them is difficult, making accurate demand forecasting critical for maximizing profits. The authors propose a new approach for optimizing pricing and replenishment decisions in vegetable departments using LSTM and genetic algorithms.
We think this is not the genuine reviewer’s comment, as we suspect this point could have been included

Round 2
Reviewer 2 Report
Comments and Suggestions for Authors
I apologise for the 13th comment. I came from a different paper.
Regarding the rest of my suggestions. I can tell the authors answered logically and I understand their motivation. Nonetheless, I cannot tell the same for the modifications. The authors merely modified 5 lines about what I suggested. I cannot modify then my decision.
The authors ar asked at least to compare with simpler models as suggested. Otherwise the results and conclusions are not fair.
Comments on the Quality of English LanguageAs the authors barely modified the paper, the same typos remain.
Author Response
We would like to present our response to the second round of comments offered by reviewer number 2. The comment in question was the following:
I apologise for the 13th comment. I came from a different paper.
Regarding the rest of my suggestions. I can tell the authors answered logically and I understand their motivation. Nonetheless, I cannot tell the same for the modifications. The authors merely modified 5 lines about what I suggested. I cannot modify then my decision.
The authors ar asked at least to compare with simpler models as suggested. Otherwise the results and conclusions are not fair.
Response 1:
We run the examples presented in the manuscript using the simpler methods that the reviewer desired. Namely:
- We added a grid search over different possible hyperparameter combinations for single decision trees to the comparison. Note that in previous manuscript, Random forest models were explored.
- We added a grid search over possible simple neural network architectures, including networks with a single neuron, a single layer and two layers, of different sizes, as well as different combinations of hyperparameters.
Please find the hyperparameter grids for both methods in the updated Table 1 on page 16.
The additional results highlight how in the explored situations simple models do not perform as good as their more complex counterparts (respectively, Random Forest and Deep Learning architectures).
The addition of more methods unfortunately made the figures significantly less clear to parse and understand. Therefore, we opted for:
- splitting each figure into two figures: one illustrating a comparison with more classical ML models and another one with deep learning models
- switching from the previous error bar visualization to highlighting the standard deviation using shaded area.
We believe those changes result in a significantly clearer exposition.
Furthermore, we also included in each figure some new plots indicating the difference in performance between the training set and test set. The updated figures clarify how the performance of DL models on the test set is not due to lack of their ability to fit the training set, but rather a lack of generalization.
We hope that the proposed improvements fully address the concerns of the reviewer.
Round 3
Reviewer 2 Report
Comments and Suggestions for Authors
No comments.
Comments on the Quality of English LanguageMinor typos detected from the very first version.